# "Not gonna lie, that's a real bummer"—The Usualization of the Pragmatic Marker *not gonna lie*

Nicole Benker

Department of English and American Studies, Ludwig Maximilian University Munich, 80539 Munich, Germany;
nicole.benker@campus.lmu.de

## Abstract

This study is concerned with the formal and functional development of the pragmatic marker *not gonna lie*. It comprises a detailed investigation into the usage and development of *not gonna lie* in American English. This study shows that *not gonna lie* develops from the clause *NP BE not going to lie to NP*. From its earliest attestations onward, the marker occurs in contexts carrying face threats, which points towards face-threat mitigation as its main function. This discourse function can only be observed for variants with first-person subjects and *you* in the prepositional phrase (if present). The later omission of elements through the course of the development indicates an increase in syntactic autonomy. The remaining chunk, *not gonna lie*, leaves little room for variability and is dominated by its discursive function. The findings are interpreted through the lens of usualization as described in the Entrenchment-and-Conventionalization Model. This dynamic, usage-based and cognitive model of language use and change lends itself to providing a fine-grained description and explanation of the grammaticalization-like processes observed in this case study.

**Keywords:** pragmatic markers; grammaticalization; construction grammar

## 1. Introduction

This paper is concerned with the development of the pragmatic marker *not gonna lie* and its variants from the perspective of usage-based Construction Grammar. A prototypical pragmatic marker is usually defined as a

> "phonologically short item that is not syntactically connected to the rest of the clause (i.e., is parenthetical), and has little or no referential meaning but serves pragmatic or procedural purposes. Prototypical pragmatic markers in Present-day English include one-word inserts such as *right*, *well*, *okay*, or *now* as well as phrases such as *and things like that* or *sort of*" (Brinton, 2008, p. 1).

It Is the "one-word inserts" and short phrasal markers that have received most attention (both theoretical and empirical) so far, e.g., comment clauses, like *I think* and *I mean* (e.g., Aijmer, 1997; Brinton, 2008; Dehé & Wichmann, 2010; Heine & Kaltenböck, 2021; Heine et al., 2021; Kaltenböck, 2010; Kärkkäinen, 2003; Van Bogaert, 2011), highly frequent multifunctional items, such as *like, so, well*, etc. (e.g., Beeching, 2015; D'Arcy, 2017; Fraser, 1990, 1999, 2009; Müller, 2005; Schiffrin, 1987; Tagliamonte, 2005) or discourse-structuring markers, like *after all, by the way* or *anyway* (e.g., Haselow, 2015; Heine et al., 2021; Traugott, 2022a, 2022b).

From studies like these, several key findings can be gleaned. Pragmatic markers tend to come from ordinary lexical items or compositional clauses with propositional meanings that develop into textual, discourse-oriented and/or interpersonal functions. The source lexemes are semantically bleached and phonetically reduced. Pragmatic markers are grammatically optional but, nonetheless, serve important pragmatic functions. Most pragmatic markers are multifunctional. They tend to be placed clause/utterance-initially but are positionally flexible and may also occur clause/utterance-finally or medially. Furthermore, their functional scope typically extends over larger chunks of discourse, not just individual lexical items (cf. Brinton, 2017, pp. 2–9, for a more detailed summary of features and functions of pragmatic markers).

Less attention has been paid to less prototypical, longer markers of clausal origin. These pragmatic markers develop from fully compositional clauses to (formulaic) pragmatic markers. For example, *needless to say* developed from the clause *it is needless to say that X* (Schmid, 2020) or *just saying* from *I am just saying X* (Brinton, 2017). Thus, the main goal of this paper is to add to the existing body of work on clausal pragmatic markers by tracing the formal and functional development of *not gonna lie* from the compositional clause *NP BE not going to lie to NP*, as in, e.g.,

1.    "Look, Don. **I'm not going to lie to you**. I'm not in great financial shape [. . .]" (1992, *Corpus of Contemporary American English*, TV/MOV).
2.    A: "I need a roommate if you wanna crash."
      B: "Hmm. **Not gonna lie**. The subway is cleaner than your couch." (2011, TV/MOV).

Previous work has often tried to explain the development of pragmatic markers through the lens of different grammaticalization frameworks (e.g., Brinton, 2008; Diewald, 2011; Heine, 2013; Heine & Kaltenböck, 2021; Hopper, 1991; Traugott, 1995; Traugott & Dasher, 2002; Zeschel et al., 2025). The reason for this approach is that the development of pragmatic markers shows some similarities to prototypical grammaticalization processes, chiefly, desemanticization and phonetic erosion (e.g., Heine & Kuteva, 2007; Hopper, 1991; Lehmann, 2015). Especially popular for the analysis of pragmatic markers from the point of view of grammaticalization were and are Hopper's (1991) principles of grammaticalization: layering, divergence, specialization, persistence and de-categorialization (for how these principles can be applied to pragmatic markers, cf. Brinton, 2017; Mroczynski, 2012).

Other characteristics of grammaticalization, such as cliticization, decrease in structural scope and integration into an inflectional paradigm (e.g., Brinton, 2017; Diewald & Smirnova, 2012; Traugott, 2022a), are not typically associated with the development of pragmatic markers. Moreover, their (sometimes) rapid development (Heine, 2013; Heine et al., 2012; Kaltenböck et al., 2011) and their doubtful status within the grammar of a language (Hopper, 1991) are further arguments against the use of grammaticalization frameworks. Therefore, attempts have been made to extend grammaticalization, e.g., by distinguishing between expansive and reductive grammaticalization (Traugott & Trousdale, 2013), by suggesting additional processes in combination with grammaticalization, e.g., cooptation (Heine, 2018; Kaltenböck et al., 2011), by using lexicalization as a framework instead (Berry, 2018; Wischer, 2000) or by developing other frameworks, for example, pragmaticalization (e.g., Claridge & Arnovick, 2010; Ermann & Kotsinas, 1993; Mroczynski, 2012, 2024) or constructionalization (Traugott, 2022a, 2022b; Traugott & Trousdale, 2013).

Because, so far, no consensus has been reached on which theoretical framework is most suitable for the study of pragmatic markers, a secondary aim of this paper is to test if the Entrenchment-and-Conventionalization Model (EC-Model, Schmid, 2020) can be applied to the development of pragmatic markers. Using the EC-Model, the present paper constitutes a detailed corpus study tracing the formal and functional development of the pragmatic marker *not gonna lie* in the *Corpus of Contemporary American English* (COCA, Davies, 2008–).

The EC-Model is a recent, usage-based and dynamic model of language use and language change broadly situated within the area of Diachronic Construction Grammar. Therefore, the model works within a specific set of basic assumptions about language (Schmid, 2020, pp. 10–11): Language is emergent, i.e., it arises through language use. Language is non-modular, i.e., there is no sharp distinction between grammar and lexicon. Language is domain-general, i.e., the basic cognitive processes that govern other higher cognitive functions, such as generalization, categorization, etc., also govern language. Language is a dynamic adaptive system that is continuously shaped by language users.

The basic linguistic units in the EC-Model are utterance types. *Utterance types* are equivalent to constructions, i.e., form–meaning pairings. Schmid (2020) does not use the term *construction* because of a slight terminological imprecision: Goldberg (1995, 2006, 2019) uses the term *construction* to refer to a cognitive entity, while Croft (2001) uses the term to refer to form–meaning pairings as used in social interaction. The EC-Model describes the dynamic interplay of both of these concepts: the cognitive representation of language in the minds of individual speakers on the side of entrenchment and the social expression of language in a speech community on the side of conventionalization, which is why Schmid (2020) opts to use the more general term *utterance type* instead (Schmid, 2020, p. 11). Furthermore, the term *utterance type* includes graphemes and phonemes, which are not generally regarded as constructions in Construction Grammar.

Entrenchment and conventionalization interact through language use. Conventionalization of utterance types is mediated on six "dimensions of conformity" (Schmid, 2020, p. 96): onomasiological, semasiological, syntagmatic, cotextual, contextual and community-related conformity. Not all of these dimensions play a major role for all utterance types; rather, certain utterance types exhibit certain conformity profiles, e.g., the conventionality of function words is mostly based on syntagmatic conformity, while the conventionality of deictic expressions is based on contextual conformity. Conventionalization is driven by two subprocesses, usualization and diffusion: "Usualization mainly affects the form-related and meaning-related dimensions [. . .]. In contrast, diffusion mainly concerns the situational and community-related dimensions of the conventionality of utterance types" (Schmid, 2020, p. 179).

Since the present paper deals with alternative theories of grammaticalization, the focus will be on usualization, which, in the context of language change, subsumes concepts such as grammaticalization, pragmaticalization, lexicalization, etc. (Schmid, 2020, p. 150). "The process of usualization establishes [. . .] and continually sustains and adapts conventionalized utterance types as [. . .] regularities of behaviour among the members of a community" (Schmid, 2020, pp. 92–93) and ". . .[usualization] controls the four cornerstones of structure: meaning, linearity, opposition and context" (Schmid, 2020, p. 93) through four subprocesses: symbolization, syntagmaticalization, paradigmaticalization and contextualization (Schmid, 2020, p. 127):

- Symbolization is concerned with how form–meaning mappings are established, preserved and, if necessary, adjusted;
- Syntagmaticalization is concerned with how linguistic elements can be sequentially arranged and combined in a linear fashion;
- Paradigmaticalization is concerned with the organization of linguistic competitors in onomasiological and semasiological networks of opposition and contrast;
- Contextualization is concerned with the usage context (incl. genre and register and situation) of utterance types.

How usualization drives language change is exemplified in Schmid (2020) using the expression *needless to say*, which developed from the "fully compositional comment clause with propositional meaning, [*it is needless to say that X*], to a fixed expression with a

metacommunicative function similar to that of pragmatic markers" (Schmid, 2020, p. 151). Syntagmaticalization leads to a rise in syntagmatic conformity, i.e., *needless* is an increasingly good predictor of *say* (Schmid, 2020, p. 151). Externally, the marker "no longer relies on syntagmatic conformity" (Schmid, 2020, p. 152), which leads to positional flexibility of the marker, but internally to the loss of paradigmatic variability in the individual parts. Because of this decrease in external syntagmatic conformity (i.e., the grammaticality of the utterance within which the chunk is placed) and the increase in internal syntagmatic conformity (i.e., coalescence of the individual elements), function words are no longer necessary to "support the conventionality of the pattern and can, therefore, be omitted" (Schmid, 2020, p. 152).

Paradigmaticalization and symbolization lead to a decrease in symbolic associations between the form and meaning of the individual items and, in turn, to a rise in the symbolic conformity of the expression as a whole, as "individual components of the sequence undergo the process of 'deparadigmaticalization' [...], whereas the whole sequence enters into new oppositions and therefore becomes paradigmaticalized" (Schmid, 2020, p. 152). On the one hand, this means that the system of functionally similar pragmatic markers already in place in the speech community and in the individual's mind is rearranged (through, e.g., semantic narrowing of other markers) to accommodate the addition of *needless to say*. On the other hand, it is no longer possible to replace *say* with a semantically related lexeme, such as *speak* or *tell*, with which *say* might be in competition in other contexts, because *say* is no longer used to denote 'verbally expressing something'. Lastly, through contextualization, the whole sequence becomes pragmaticalized. "This means that cotextual and contextual factors begin to dominate the choice of the whole expression in the pursuit of the communicative goal of emphasizing something in spite of its being obvious to the speaker and hearer" (Schmid, 2020, p. 152).

The EC-Model does not claim to reinvent grammaticalization, pragmaticalization, constructionalization or similar processes but rather aims to integrate these processes into a dynamic model of language and cognition. What sets the EC-Model apart from other cognitive models (e.g., Heine, 2019) is that it describes both the distinction and the interaction of language as it is used in a speech community (conventionalization) and language as it is used by the individual (entrenchment), which makes it an attractive approach to study linguistic changes, such as the development of pragmatic markers.

Because *not gonna lie* is a fairly recently developed pragmatic marker, it can neither be found in traditional dictionaries nor in previous scholarship. There are, however, crowd-sourced online dictionaries that offer definitions and usage examples. According to *Urban Dictionary*, *(I'm) not gonna lie* is "synonymous with: *to tell you the truth, honestly, actually, as a matter of fact, in fact, truthfully*" (*Urban Dictionary*, 1999–, s.v. *I'm not gonna lie*). Furthermore, "it makes your statement more valid/less offensive." (*Urban Dictionary*, 1999–, s.v. *ngl*), which indicates a mitigating function. While there is no previous scientific work on *not gonna lie* specifically, there are other similar expressions using lexemes from the source domain TRUTH/FACT that have received attention in the past. The domain of TRUTH/FACT as a source for stance adverbials was previously explored by Biber and Finegan (1988), who identify two groups of adverbials that draw from the domain of TRUTH/FACT: the *honestly* adverbials (also including, e.g., *candidly*, *frankly*, etc.) and the *actually* adverbials (also including, e.g., *in fact*, *in actuality*, *as a matter of fact*). Especially the *actually* adverbials, which tend to convey counterexpectancy (e.g., Aijmer, 2002, 2013; Aijmer & Simon-Vandenbergen, 2004; Álvarez-Gil, 2017; Beekhuizen et al., 2024; Biber & Finegan, 1988; Chafe, 1986; Oh, 2000; Simon-Vandenbergen, 2014; Simon-Vandenbergen & Aijmer, 2002, 2007; Simon-Vandenbergen & Willems, 2011; Usonienė et al., 2015; Vision, 2008), and TRUTH-intensifiers,

like *really* and *truly*, which are commonly used emphatically (Beekhuizen et al., 2024; Defour, 2012; Paradis, 2003; Swan, 1988b; Vision, 2008), have been studied extensively.

Lenker (2010) suggests that the TRUTH/FACT adverbials she investigated, e.g., *truly*, *soothly* or *in fact*, often gain concessive or contrastive function as conversational implicature, while their conventionalized function is TRANSITION (Lenker, 2010, p. 114). Lenker further posits that these functions are a result of adherence to Gricean maxims. Lying or stating untruths flouts the Maxim of Quality, and thus, explicitly stating that an utterance is truthful is only acceptable in contexts where the veracity of a statement could be called into question, for example, in negative contexts (Lenker, 2010, p. 129). Using an adverbial with lexical items from the TRUTH/FACT domain thus leads the interlocutor to interpret the adverbial as either having concessive or epistemic function, which, in turn, implies an increase in subjectification (Lenker, 2010, pp. 129–130).

The adverbial TRUTH markers of the type *honestly* and *frankly* have so far received less attention (Edwards & Fasulo, 2006; Tseronis, 2011). These markers are commonly classified as speech-act or illocutionary adverbs (Berry, 2018; Ifantidou, 2008; Keizer, 2018; Tseronis, 2011) or style disjuncts (Quirk et al., 1985), which have been described as having intersubjective and politeness/mitigating function (Berry, 2018, p. 150; Tseronis, 2011, p. 478). *Frankly* is used to "express an opinion of the speaker, one he/she expects other people to disagree with" (Keizer, 2018, p. 70) and to acknowledge "that other views may exist with which the particular standpoint clashes and thereby seeks to reinforce the justificatory force of the arguments adduced in support of it" (Tseronis, 2011, p. 483) while emphasizing "the speaker's sincerity and cooperativeness" (Tseronis, 2011, p. 484). Keizer (2018) further identifies concessive function and that "the use of *frankly* typically involves a certain degree of counterexpectancy" (Keizer, 2018, p. 69), while Tseronis (2011) explicitly excludes signaling unexpectedness from the functions of *frankly* and firmly places this function with the ACT and FACT adverbials discussed above. Furthermore, Keizer (2018) posits a kind of "hortative" function, i.e., an encouragement towards the hearer to agree with the speaker's statement (Keizer, 2018, p. 80). Regarding the development of *frankly* and *honestly*, Berry (2018) argues that they arise through the lexicalization of clauses, such as *I tell you frankly*. Because *frankly* is so strongly associated with "the act of speech [...] it metonymically contains the omitted interlocutors (the first- and second-person deictics) and the speech-act adverb" (Berry, 2018, p. 145). The only element that remains is the most salient and significant, namely, *frankly* (Berry, 2018, p. 145; cf. also Swan, 1988b for similar conclusions, though not through the lens of lexicalization).

Even less work has been dedicated to clausal TRUTH markers. Edwards and Fasulo (2006) more broadly investigated "honesty phrases", i.e., pragmatic expressions such as *to be honest*, *if I am being honest*, etc., in talk-in interaction and found that they tend to be used to introduce dispreferred answers and appear in contexts of "non-answers to expectably answerable questions in confessions of failed incumbency to perform a service; and in generally negative, delicately broached assessments of persons (particularly spouses and work colleagues) known to both parties" (Edwards & Fasulo, 2006, p. 371).

While not drawing upon the prototypical TRUTH/FACT lexemes, *I (must) admit* and *admittedly*, two epistemic parentheticals (Brinton, 2017, p. 177), are rather well-studied markers with similar functions to what the *Urban Dictionary* entries suggest for *(I'm) not gonna lie* and what previous scholarship has identified for *frankly* and *honestly*. Fraser (1975) states that *must* in *I must admit* implies that the user "would like to be relieved of at least some of the onus of the consequences, such as not antagonizing the hearer or countering the hearer's view" (Fraser, 1975, p. 196). Furthermore, Swan (1988a, p. 45) claims that *admittedly* is used as a hearer-based hedge. Similarly, Brinton (2017) connects the use of certain markers to politeness functions, specifically politeness in the sense of Brown and

Levinson ([1987]). Brinton ([2017]) states that the expressions *if I may say so* and *for what it's worth* are used to mitigate attacks against the hearer's negative face because both markers tend to be used when the speaker anticipates that the hearer might disagree with their opinion or statement (Brinton, [2017], pp. 230–231) and thus have a hedging and politeness function, again reminiscent of the uses cited by the *Urban Dictionary* users.

Similar analyses have been conducted for German markers with similar meaning and function that use lexemes from the domain of TRUTH/FACT, i.e., *ehrlich/offen gesagt* (lit. 'honestly/openly said') (Hagemann, [1997]; Niehüser, [1987]; Rolf, [1994]; Zeschel et al., [2025]) and *wenn ich ehrlich bin* ('if I am being honest') (Günthner, [2024]). Niehüser ([1987]) identifies three functions of *ehrlich gesagt*: (1) announcing a violation of conversational/politeness norms, (2) correcting a previously made, dishonest statement and (3) affirming a statement that is unexpected or not credible (Niehüser, [1987], p. 181). For *um die Wahrheit zu sagen*, he additionally identifies the function of resolving misunderstandings in the context of irony, i.e., making clear that a previous utterance was ironic by emphasizing the truthfulness of the following statement (Niehüser, [1987], p. 180). While Hagemann ([1997]), Rolf ([1994]) and Niehüser ([1987]) relate these expressions primarily to Grice's cooperative principle, Zeschel et al. ([2025]), in an analysis concerned with *ehrlich gesagt*, relate these functions to face-work according to Goffmann ([1967]) and Brown and Levinson ([1987]). They state that *ehrlich gesagt* is used in contexts that threaten the speaker's face but also in ones that threaten the hearer's face (Zeschel et al., [2025], p. 224). In their data, *ehrlich gesagt* was usually used utterance-initially and is used to both prepare the addressee for an upcoming face-threatening act and mitigate said threat (Zeschel et al., [2025], p. 225). Because the expression's source material is taken from the domain of TRUTH, the FTA is licensed by the Maxim of Quality (Zeschel et al., [2025], p. 224). Zeschel et al. ([2025]) primarily relate face-work to functions (1) and (2) in Niehüser ([1987]), while they classify function (3) as a non-face-related function. However, the example they use to illustrate these types of functions ("Ja, ich bin ehrlich gesagt happy, daß mir der erste Absprung so gut gelungen ist. [...]" ['Yes, to be honest, I am happy that my first jump was so successful'] (Zeschel et al., [2025], p. 225)) could be interpreted as boasting, which constitutes an FTA against the hearer's positive face according to Brown and Levinson ([1987], p. 67).

Based on these previous studies, the importance of mitigation/politeness functions, as described in Grice ([1975]) and Brown and Levinson ([1987]), has become clear. However, all of the studies analyze politeness functions in a qualitative manner and mainly explore the function or functional development of the respective pragmatic markers. The present study will explore both the formal and functional changes in *(I'm) not gonna lie (to you)* and use these theories of politeness as the basis for a detailed quantitative analysis of how *not gonna lie* and its variants are used.

Thus, this paper aims to show that during its development from the clause *I'm not going to lie to you*, *not gonna lie* becomes strongly associated with a mitigating function (symbolization) in the context of face-threatening acts (contextualization). This means semantic bleaching of the component parts of the expression and loss of compositionality. Through syntagmaticalization, the clause *I'm not going to lie to you* should become fixed and invariant. The loss of compositionality should result in the omission of (grammatical) elements because the expression no longer relies on the rules of grammar and, further, the loss of paradigmatic choices within the expression ("deparadigmaticalization"). This should also lead to positional flexibility. Theoretically, the expression should then enter into new systems of opposition, but this cannot be properly investigated without investigating other pragmatic markers with similar functions as well.

## 2. Materials and Methods

The corpus used for this study is a large-scale corpus featuring American English data: the *Corpus of Contemporary American English* (COCA, Davies, 2008–). COCA contains about one billion words from 1990 to 2019, which constitutes the relevant time frame of the development of *not gonna lie*. Furthermore, *not gonna lie* is rather infrequent, which necessitates a large dataset. COCA is balanced by year and by the following genres: spoken, fiction, popular magazines, newspapers, academic texts, TV and movie subtitles. The later additions of blogs and other websites contain material only from the year 2012 and thus do not allow for a diachronic investigation, which is why these two genres were excluded from this analysis. Except for source genre and year, no other metadata is available for COCA.

The relevant attestations were extracted using the search query * * *not* * * *lie* * *, which allows for variation in the subject slot (NP1), the verb slot (forms of *to be*), the modal slot (*going to* and *gonna*), the preposition slot (PREP) and the object slot (NP2), to find tokens with the structure *NP1 BE not GOING TO lie PREP NP2*, e.g., *I'm not going to lie to you.* Additionally, the search query * * *not lie* * * (and *n't* for contracted forms) was used to allow for other modal verbs, e.g., variants with the structure *NP1 BE MODAL not lie PREP NP2*, such as *I cannot lie to you* or *I won't lie to you*. This approach allows for the study of more compositional and less compositional instances of the expression.

These search queries yielded 1691 attestations in COCA, of which 1299 belonged to the relevant syntagmatic strings. "Relevant" in this context meant that *lie* was not used in the sense "move into/be in a horizontal position" (e.g., *He's not going to lie down*), that the attestations were actually variants of the target string (and not, e.g., *it is not permitted to lie*) and that *lie* was not coordinated with another verb (e.g., *I'm not going to lie and say that . . .*). Duplicates were removed. An attestation was classified as a duplicate if the same search result occurred more than once in the data (i.e., same year, same source text). If a quote was used multiple times in the data in different texts, these attestations were not classified as duplicates and therefore not removed.

The tokens were then tagged for "variant" (e.g., *NP BE not GOING TO lie*, *NP cannot lie about NP* etc.), the fillers for the open slots (e.g., *I*, *she*, etc., for NP1, *to*, *about*, *on*, etc., for PREP, *can*, *will*, *going to*, etc., for the modal and *you*, *them*, *her*, etc., for NP2), the tense of *to be* and their function (literal, i.e., *lie* is used in the sense 'not telling the truth' vs. epistemic, i.e., the expression is used with pragmatic function, following Brinton (2017, 230f.) and Traugott (2003, p. 128)). In the case of epistemic function, their "position" in reference to the matrix clause (anaphoric, cataphoric, i.e., if the token is used to modify the previous or the following statement) was analyzed. This analysis corresponds to analyses of pragmatic markers in the left and right periphery (e.g., Beeching & Detges, 2014). Since many of the attestations are main clauses in their own right and sentence boundaries are unclear in large parts of the data, it is difficult to determine peripheries. Moreover, many attestations appear to be in an intermediate state between an independent clause and a pragmatic marker. Thus, following Blakemore (1987) and Schiffrin (1987), I made the decision to instead use the terms *anaphoric* and *cataphoric reference*, which more accurately represent these concepts for clause-like structures, such as the one discussed in the present paper.

Furthermore, attestations with either *going to* or *gonna* in the modal slot were combined because of the mixed nature of the material. In the spoken as well as movie/TV subtitle sections, it is not possible to verify the actual form that was being produced because no audio files are available for the corpus. Since *going to* and *gonna* are phonetically rather similar and can be difficult to tell apart in connected speech, they are treated as one form in the present paper. Considering that *will not* and *won't*, as well as *cannot* and *can't*, are much more easily distinguished auditorily, I made the decision to treat attestations with this variation as separate variants.

In concrete terms, the analysis looks like the following:

3.   "When he asks me, **I'm not going to lie**," she said. (2017, MAG).

In (3), the variant is *NP BE not GOING TO lie*, the subject slot is filled with *I*, the preposition and the object slot are omitted. The example is in present tense, and the phrase is used in a literal sense. Since the meaning is literal, reference to other parts of discourse was not tagged.

As a second step, all tokens tagged as "epistemic" in the category "function" were analyzed according to their usage context (842 attestations). Following Brinton's (2017) qualitative analysis of *for what it's worth* and *if I may say so*, the presence or absence of (potential) face-threatening acts (FTAs) in the context was investigated. When a possible FTA was identified, the FTA was further specified according to the four categories laid out in Brown and Levinson (1987, pp. 65–68) (listed here is an abbreviated version with only the relevant categories for the present data):

- **FTA is committed against the hearer's (H's) negative face:** suggestions, advice; warnings; compliments, expressions of envy or admiration; expressions of strong negative emotions towards the H.
- **Against the H's positive face:** expressions of disapproval, criticism, contempt or ridicule, complaints and reprimands, accusations, insults; contradictions or disagreements, challenges; expressions of violent (out-of-control) emotions; irreverence, mention of taboo topics; bringing of bad news about the H or good news (boasting) about the S.
- **Against the speaker's (S's) negative face:** excuses.
- **Against the S's positive face:** confessions, admissions of guilt or responsibility; emotion leakage, non-control of laughter or tears.

For example, in (4), the interviewee is bragging about his skills and thus commits an FTA against the hearer's positive face. In (5), the speaker offers advice and thus commits an FTA against the hearer's negative face. Lastly, (6) could be potentially added to two categories: The utterance could be interpreted as an attack against the hearer's positive face if the reference to sex work is regarded as the mention of a taboo topic. It is also a personal confession of the speaker, which constitutes an FTA against the speaker's positive face. Cases like this are not uncommon, especially the constellation of a confession that also mentions taboo topics. In these cases, the attestation was classified according to the "stronger" FTA, i.e., in the case of (6), the personal admission of being a sex worker was regarded as a stronger FTA than the mention of sex work in general (and, therefore, the FTA against the hearer).

4.   "I'm not cocky, but **I'm not going to lie to you**," Wheatley said last week. "I'm a hell of a player." (1997, NEWS).
5.   "There's always a risk. **I won't lie to you**. But if she were my mother, I'd take it." (2004, MOV).
6.   "I just... Well, **I'm not gonna lie to you**. I'm a high-end gigolo." (2018, MOV).

Cases where no FTA could be identified or the context was unclear were tagged as such.

To more specifically evaluate the degree of syntagmaticalization, the forward transition probabilities between the individual elements of each variant were calculated per year, which represents the probability with which earlier elements are followed by later elements, i.e., the degree to which earlier elements are predictors of later elements (e.g., the likelihood that *pay* is followed by *attention*). This was calculated using the equation $p(w_n w_{n+1})/p(w_n)$ (e.g., Pelucchi et al., 2009), in which $p$ is the frequency, $w_n$ is the first element in a variant (e.g., *I*) and $w_{n+1}$ is the second element (e.g., *am*). Analogously to *going to/gonna*, *am* and

*'m* are treated as the same lexeme. Therefore, their frequencies were added together in the calculation of the transition probabilities. Furthermore, the contracted forms of negated modals (i.e., *can't* and *won't*) were treated as one element in these calculations. For example, the probability of *I* being followed by *can't* in the year 1990 in COCA was calculated by dividing the frequency of *I can't* (which occurs 4157 times in 1990) by the frequency of *I* (which occurs 312,660 times in 1990), which results in a forward transition probability of 0.013.

Because the texts in COCA only sparingly conform to punctuation rules and do not contain standardized tags for the beginning of clauses, it is unfortunately not possible to accurately calculate forward transition probabilities of clause-initial elements. This means that it is not possible to calculate the probabilities for the variants without a subject slot (e.g., *can't lie*) because all attestations with subjects would be included in the numbers automatically extracted from COCA and would require manual exclusion, which, for such high-frequency items like *not* or *can't*, is not feasible.

## 3. Results

The results are presented in three sections. Sections 3.1 and 3.2 constitute investigations into syntagmaticalization and "deparadigmaticalization", i.e., its loss of compositionality, increase in formulaicity and increase in positional flexibility. Therefore, Section 3.1 describes the variants found in COCA from a formal perspective, i.e., the number of available slots, the possible slot fillers and transition probabilities, while Section 3.2 presents the position analysis. Section 3.3 is dedicated to the functional analysis using the politeness theory by Brown and Levinson (1987) to evaluate the processes of symbolization and contextualization.

### 3.1. Form

Overall, 1299 relevant attestations were extracted from COCA. In total, 26 variants were extracted, of which Figure 1 depicts the most frequent variants: *NP BE not GOING TO lie to NP* (282 attestations), *NP BE not GOING TO lie* (327) and *not GOING To lie* (29). Other, more frequent variants include *NP can't lie to NP* (140), *NP can't lie* (156), *NP won't lie to NP* (106) and *NP won't lie* (89). An overview of all variants and their frequencies can be found in Appendix A. From 1990 to 2019, there is a visible increase in the use of the target phrase and its variants, from 16 attestations in 1990 to 81 attestations in 2019. Note that at this stage, all subvariants of these forms are included, i.e., including the *going to/gonna* variation in the *going-to* variants and all slot fillers for the noun phrases of all variants. Two slots are already specified at this stage of the analysis: the prepositional slot and the modal slot. This choice was made because the preposition *to* is generally the most frequent one after the verb *lie* in the sense 'not telling the truth' regardless of the composition of the rest of the phrase. This is also reflected in the results. All variants with *lie to* are more frequent than the ones with other possible prepositions (*about*, *for*). Because variants with different modals are quite frequent, these will be regarded separately in the following.

More than half of all attestations (669 or 52%) feature *going to/gonna* in the modal slot, which makes them the largest group of variants. Figure 2 depicts the diachronic development of the most frequent *going-to* variants. From 1990 to 2019, there is a stark increase in the use of this group of variants, similar to the overall increase in all variants. Figure 2 only shows the most common variants: *NP BE not GOING TO lie to NP*, *NP BE not GOING TO lie* and *not GOING TO lie.* Other variants with *going to/gonna* are rare: *not GOING TO lie to NP* occurs only two times, *NP BE not GOING TO lie about NP* occurs 23 times, and *NP BE not GOING TO lie for NP* occurs 6 times. These will not be discussed further.

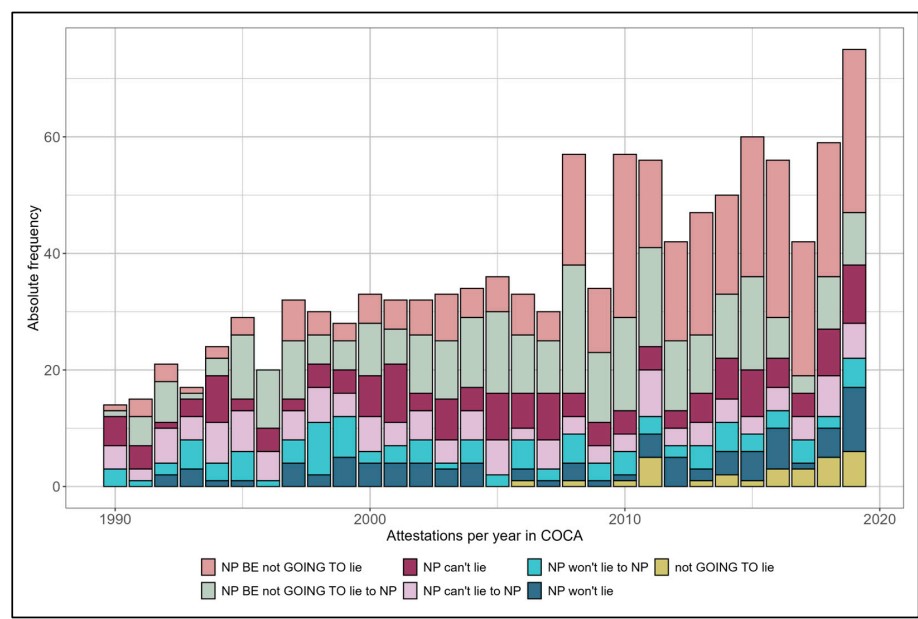

**Figure 1.** Results for the most frequent variants of the target string in COCA in absolute numbers.

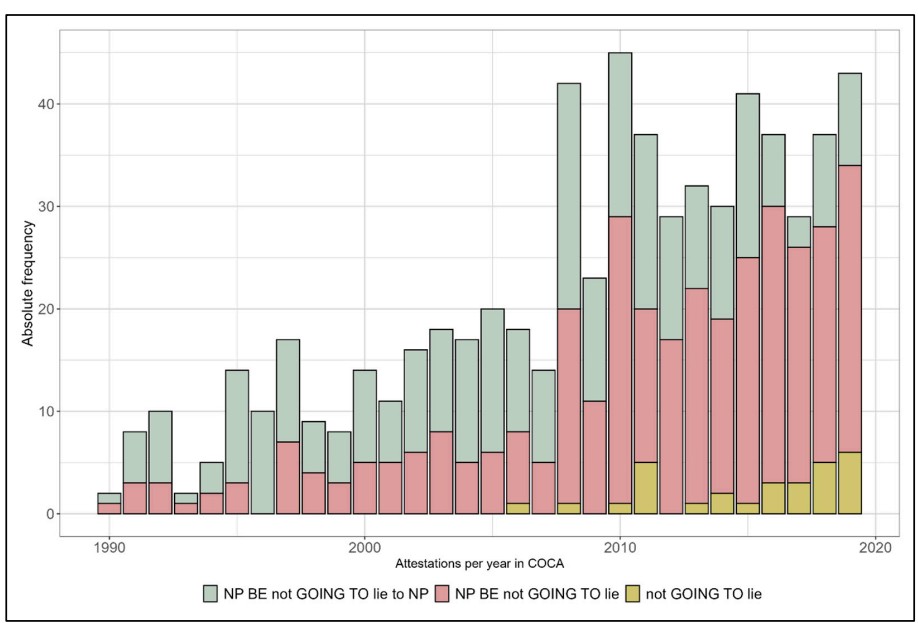

**Figure 2.** Results from COCA showing the most frequent *going-to* variants in absolute numbers.

The longest variant with the most open slots, *NP BE not GOING TO lie to NP* (282 attestations; in green), mostly appears in the form *I'm/I am not going to/gonna lie to you* (246 or 87% of these 282 attestations) and is attested from 1990 onward. However, the use of this variant does not increase over time but rather fluctuates around a mean of about 9 attestations per year. The subject slot is filled with *I* in 271 attestations (96%), the subjects in the other 11 attestations comprise *you* (7 times) and *we*, *they*, *she* and a non-pronominal NP (each 1 time). The object slot is usually filled with *you* (249 or 88%; 4 of these are rendered as *ya*); the other fillers of the object slot are a variety of pronouns (22 attestations: *her*, 5 times; *him*, 4 times; *me*, 5 times; *myself*, 1 time; *them*, 6 times; *us*, 1 time) and non-pronominal NPs (11 attestations). The second variant, *NP BE not GOING TO lie*, is missing the prepositional phrase. This variant is the most common one overall (327 attestations, in red). Like *NP BE not GOING TO lie to NP*, this one is attested from 1990 onward. Unlike the previously discussed variant, *NP BE not GOING TO lie* does increase in use from 1990. This suggests that it is mostly this variant that

drives the overall increase. The only open slot in this variant is the subject slot, which is filled with *I* about 96% (314 attestations) of the time. The other subjects are *he* (3 attestations), *they* (3 attestations), *we* (4 attestations), *you* (2 attestations) and non-pronominal NPs (1 attestation). The most reduced form of the *going-to* variants is *not GOING TO lie*, with 29 attestations, which is attested from 2006 onward. From its first attestation in 2006 to 2019, there seems to be a slight but steady increase, but since there are so few attestations, this is difficult to judge. This variant does not have open slots.

The other most frequent variants feature *can't* and *won't* in the modal slot, cf. Figure 1. Of the 1299 attestations, 322 feature the modal *can't* (~25%), while 212 feature *won't* (~16%). The non-contracted forms are much rarer: there are 16 variants with *cannot* and 38 with *will not*. Unlike the *going-to* variants, the variants with *can't* and *won't* do not show a steady increase over time but rather fluctuate, cf. Figure 3. The *can't* variants *NP can't lie to NP* (139 attestations) and *NP can't lie* (156 attestations) are both attested from 1990 onward, while *can't lie* only occurs 2 times (2012 and 2019), cf. Figure 3. Other attested *can't* variants are *can't lie to NP* (3 attestations), *NP can't lie about NP* (19 attestations) and *NP can't lie for NP* (2 attestations). In terms of slot fillers, the *can't* variants show much more variability than the *going-to* variants: For *NP can't lie to NP*, the subject slot is filled with *I* 77 times (~55%), *you* 49 times (~35%), *he* 2 times, *they* 1 time and *we* and non-pronominal NPs 5 times each. The object slot is filled with non-pronominal NPs 42 times, *you* 40 times, *me* 22 times, *him* 8 times, *them* 7 times, *her* 5 times, *us* 4 times, different reciprocal pronouns 8 times and different indefinite pronouns 3 times. The subject slot in *NP can't lie* is filled with *I* 93 times (~60%), *you* 27 times (~17%), non-pronominal NPs 10 times, *they* 8 times, *we* 6 times, *he* 5 times, *she* 4 times and *who* 3 times.

The longest *won't* variant *NP won't lie to NP* (106 attestations) is attested from 1990 onwards, *NP won't lie* (89 attestations) from 1992 onwards and *won't lie* occurs only 1 time in 2019 (cf. Figure 3). *NP won't lie to NP* does not increase over time, while *NP won't lie* does perhaps slightly increase in the 2010s, but since the numbers per year are so low, this is difficult to judge. Other attested *won't* variants are *NP won't lie about NP* and *NP won't lie for NP* (8 attestations each). In this group of variants, the longest variant is also the most frequent one, which is not the case for the *going-to* and *can't* variants. In terms of slot fillers, the *won't* variants show much less variability than the *can't* variants and behave more similarly to the *going-to* variants. The subject slot in *NP won't lie to NP* is filled with *I* 92 times (~87%), *he* 4 times, *you*, *she* and non-pronominal NPs each 2 times and *that*, *they*, *we* and *who* each 1 time. The object slot is filled with *you* 92 times (~87%), *me* 7 times, non-pronominal NPs 4 times, and *her*, *him* and *them* each 1 time. The subject slot in *NP won't lie* is filled with *I* 82 times (~92%), *they* and *we* each 2 times and *he*, *you* and non-pronominal NPs each 1 time.

In summation, the open slots in the most frequent variants (*going-to* variants) show the least variability. The *won't* variants are a little more variable than the *going-to* variants. The most variable group of variants are the *can't* variants, even though these are more frequent than the *won't* variants. Furthermore, the more reduced forms, i.e., the forms that omit the prepositional phrase, are slightly less variable in the subject slot than the longer forms with the prepositional phrase. This is the case for the *can't* and *won't* variants; for the *going-to* variants, the variability in the subject slot stays the same, which is not unexpected since the form with the PP already shows next to no variability in the subject slot. For the *going-to* and *can't* variants, the longer forms with the prepositional phrase are less frequent than the variants without the prepositional phrase. This is not true for the *won't* variants. The variants that omit both subject and prepositional phrase, however, are the least frequent across all variant groups.

As a measure of formal variability, and in order to test the degree of syntagmaticalization, the forward transition probabilities per year per variant were calculated. As shown in the previous sections, most potential slot fillers are quite rare, so the transition probabilities

were only calculated for the most common instantiation of each variant group, i.e., *I AM not GOING TO lie to you* (like in the previous sections, *AM* includes both *am* and *'m* and *GOING TO* includes both *going to* and *gonna*), *I can't lie to you* and *I won't lie to you*.

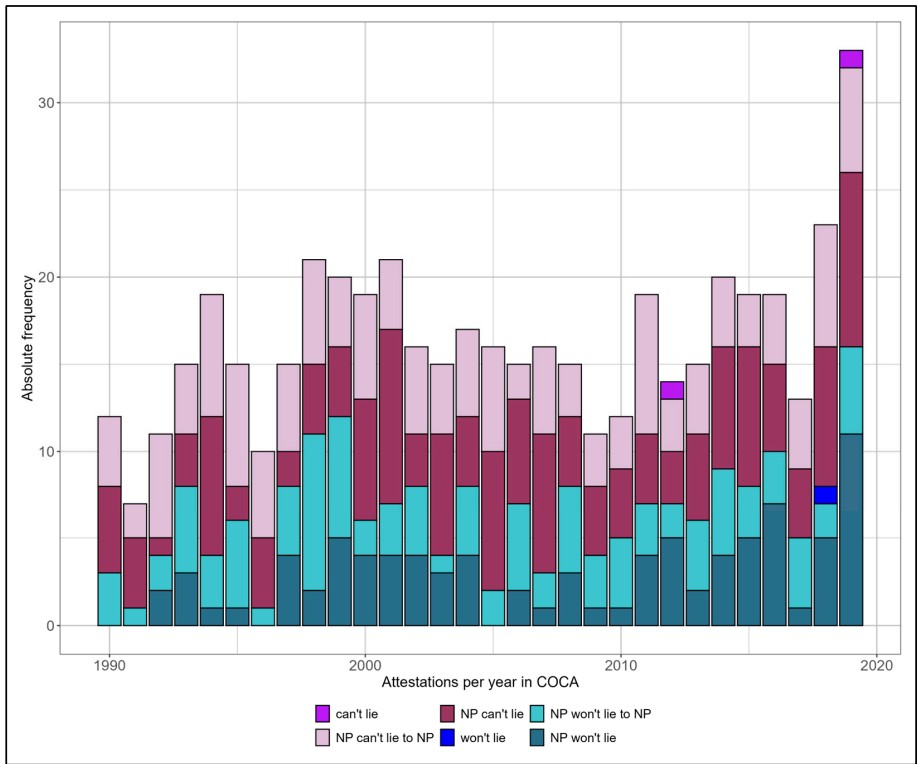

**Figure 3.** Results from COCA showing the most frequent *can't* and *won't* variants.

Figure 4 depicts the forward transition probabilities for the three markers averaged over the entire timeframe. While most of the transition probabilities are simply the result of effects that are not specific to the syntagmaticalization of these specific pragmatic markers (i.e., *lie* being often followed by *to*), some insights can be gleaned from the Figure 4. As the results from the previous section suggest, the *can't* variants are much more variable than the *going-to* variants and the *won't* variants, which is reflected in their overall much lower transition probabilities.

More insightful than the overall transition probability are the transition probabilities per year. Because most of the forward transition probabilities simply reflect the grammar of the English language and the regarded timeframe is rather short, most forward transition probabilities remain stable (e.g., the probability that I is followed by am/'m has not changed from 1990 to 2019). This section will thus only highlight the probabilities that change over time.

The first transition that shows a slight change over time is the transition *I AM not GOING TO > lie*, which mirrors the increase in the frequency of *NP not GOING TO lie* shown in the previous sections (cf. Figure 5, left panel). The other two markers show no such increase, which indicates no clear evidence for increased internal syntagmatic conformity. The transition to the preposition shows a more drastic change over time (cf. Figure 5, right panel). For all three markers, the forward transition probability to the preposition massively decreases over time. While this development is the most drastic for *I won't lie > to*, it is apparent for all three markers. In order to verify that this decrease is not a reflection of a change to the verb's valency and thus a development independent from the syntagmaticalization of the pragmatic markers, the transition probabilities for *lie > to* were calculated over time as well (cf. gray line in Figure 5, right panel). As the graph indicates,

the decrease shown for the pragmatic markers is not related to a general change in the valency of the verb *to lie*. In fact, the transition *lie > to* shows the opposite tendency, namely, a slight increase over time.

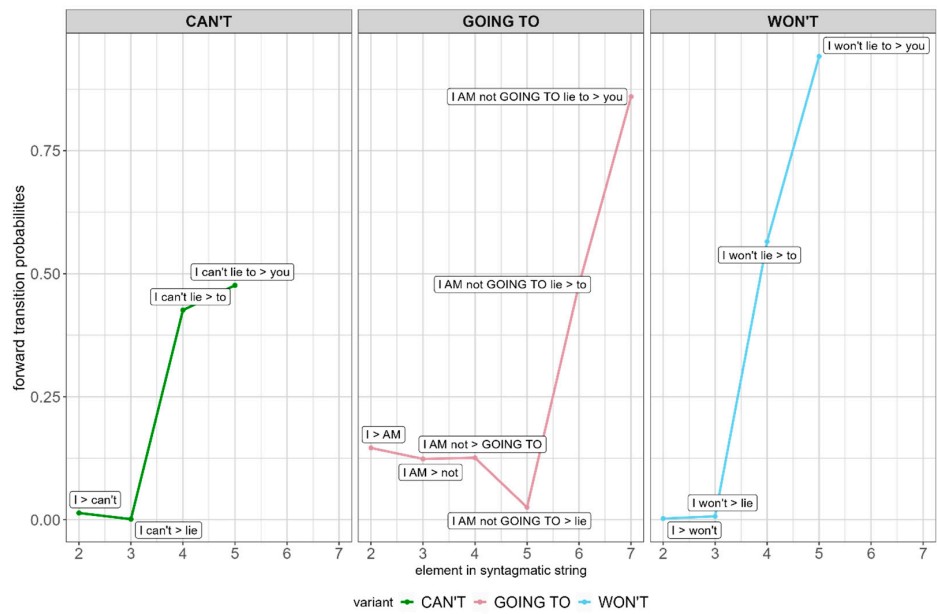

**Figure 4.** Results for the forward transition probabilities of the three variants *I can't lie to you* (green, left panel), *I AM not GOING TO lie to you* (red, middle panel) and *I won't lie to you* (blue, right panel). The *y*-axis plots the forward transition probabilities. The numbers on the *x*-axis refer to the elements in the syntagmatic string, i.e., *I* is element 1, *can't* is element 2, etc., i.e., the dot at the number 2 on the *x*-axis represents the forward transition probability from element 1 to element 2 (e.g., from *I* to *can't*).

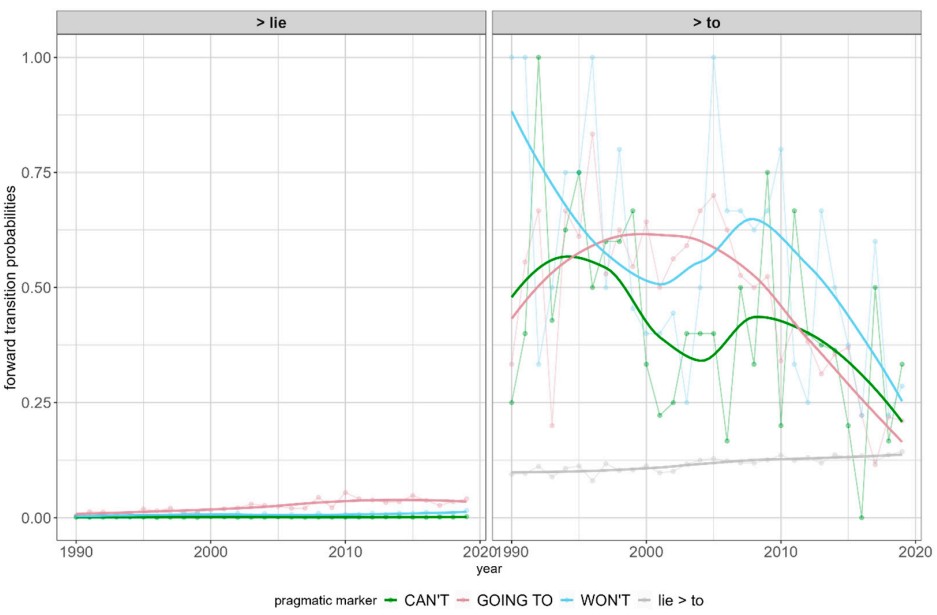

**Figure 5.** Results for the forward transition probabilities per year for all three markers. The left panel depicts the forward transition probability to the verbal element *lie* and the post-verbal preposition *to* for the pragmatic markers *I can't lie to you* (green, i.e., transition from *I can't* to *lie* and *I can't lie to to*), *I AM not GOING TO lie to you* (red) and *I won't lie to you* (blue). Additionally, the transition probabilities of the verb *lie* to the preposition *to* are plotted as a reference in the right panel (in gray). The bold lines are generated using the tidyverse smooth function, while the transparent lines depict the actual values of the transition probabilities.

### 3.2. Position

As already mentioned, instead of analyzing the position of the pragmatic markers in relation to the peripheries of a matrix clause, I opted to analyze the reference of the marker to previous or following discourse because many attestations are rendered as (orthographic) sentences in the data, i.e., occur in between periods. However, when markers are referred to as having anaphoric reference, this should be regarded as equivalent to what other studies refer to as right-peripheral/clause-final position (and cataphoric reference as left-peripheral/clause-initial). Like the functional analysis, the position analysis was only conducted on the 842 non-literal attestations. Of these 842 attestations, 663 have cataphoric reference (~79%), 123 have anaphoric reference (~15%), 8 are positioned medially (~1%) and 48 are ambiguous (~6%).

These distributions are reflected in the individual variant groups. The 559 non-literal attestations featuring *going-to* variants comprise 435 that have cataphoric reference (~78%), 86 that have anaphoric reference (~15%), 5 that are positioned medially (~1%) and 33 that have unclear reference (~6%). *NP BE not GOING TO lie to NP* (229 attestations) has cataphoric reference in 189 cases (~83%), anaphoric reference in 20 cases (~9%), medial position in 2 cases (~1%) and unclear reference in 18 cases (~8%), cf. Figure 6 and Table 1. *NP BE not GOING TO lie about NP* (8 attestations) has cataphoric reference in 3 cases and anaphoric reference in 5. *NP BE not GOING TO lie* (291 attestations) has cataphoric reference in 221 cases (~76%), anaphoric reference in 54 cases (~19%), medial position in 3 cases (~1%) and unclear reference in 13 cases (~4%). *Not GOING TO lie* (29 attestations) has cataphoric reference in 21 cases (72%), anaphoric reference in 7 cases (24%) and unclear reference in 1 case. *Not GOING TO lie to NP* (2 attestations) has cataphoric reference in one case, and the other attestation has unclear reference.

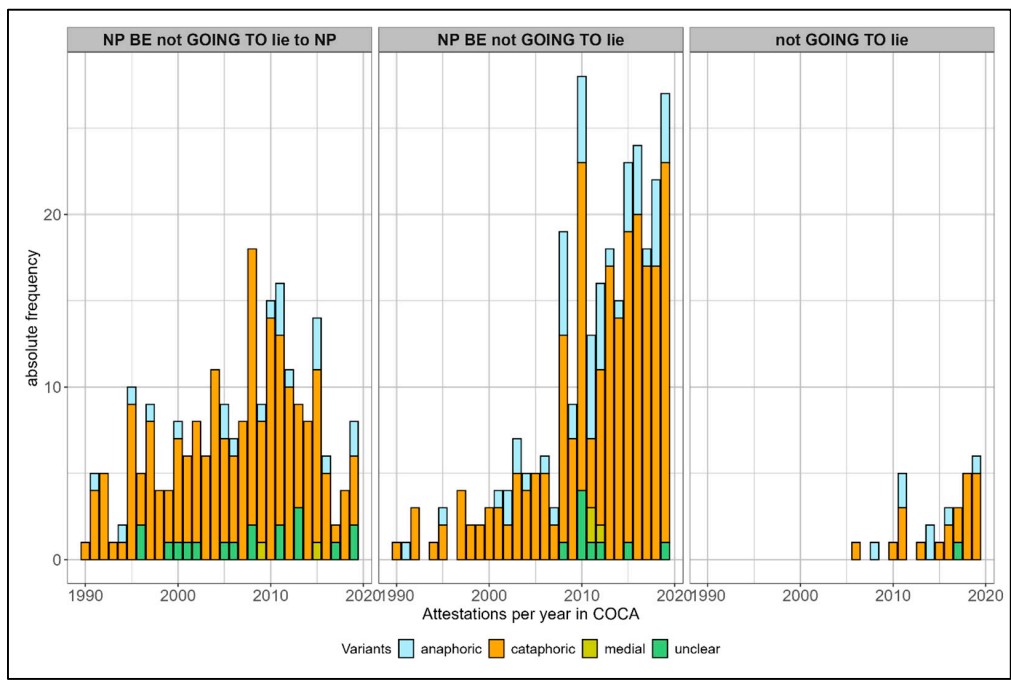

**Figure 6.** Results for reference/position analysis of *going-to* variants in COCA in absolute numbers. Anaphoric and cataphoric reference can be regarded as equivalent to right- and left-peripheral, while medial refers to clause-medial position.

Of the 92 attestations featuring *can't* variants, 71 have cataphoric reference (~77%), 10 have anaphoric reference (~11%), 2 are medially positioned (~2%) and 9 have unclear reference (~10%). *NP can't lie to NP* (30 attestations) has cataphoric reference 26 times

(~87%), anaphoric reference 1 time (~3%) and unclear reference 3 times (10%), cf. Figure 7 and Table 1. *NP can't lie* (59 attestations) has cataphoric reference 42 times (71%), anaphoric reference 9 times (15%), medial position 2 times (3%) and unclear reference 6 times (10%). Both attestations of *can't lie* have cataphoric reference.

**Table 1.** Summary of the results for the position analysis in percent relative to the number of attestations of each variant, rounded to the nearest whole number.

| Variant | Cataphoric | Anaphoric | Medial | Unclear |
|---|---|---|---|---|
| *NP BE not GOING TO lie to NP* | 83% | 9% | 1% | 8% |
| *NP BE not GOING TO lie about NP* | 38% | 63% | - | - |
| *NP BE not GOING TO lie* | 76% | 19% | 1% | 4% |
| *not GOING TO lie* | 72% | 24% | - | 3% |
| *NP can't lie to NP* | 87% | 3% | - | 10% |
| *NP can't lie* | 71% | 15% | 3% | 10% |
| *can't lie* | 100% | - | - | - |
| *NP won't lie to NP* | 92% | 8% | - | - |
| *NP won't lie about NP* | - | 100% | - | - |
| *NP won't lie* | 79% | 16% | 1% | 3% |
| *won't lie* | 100% | - | - | - |

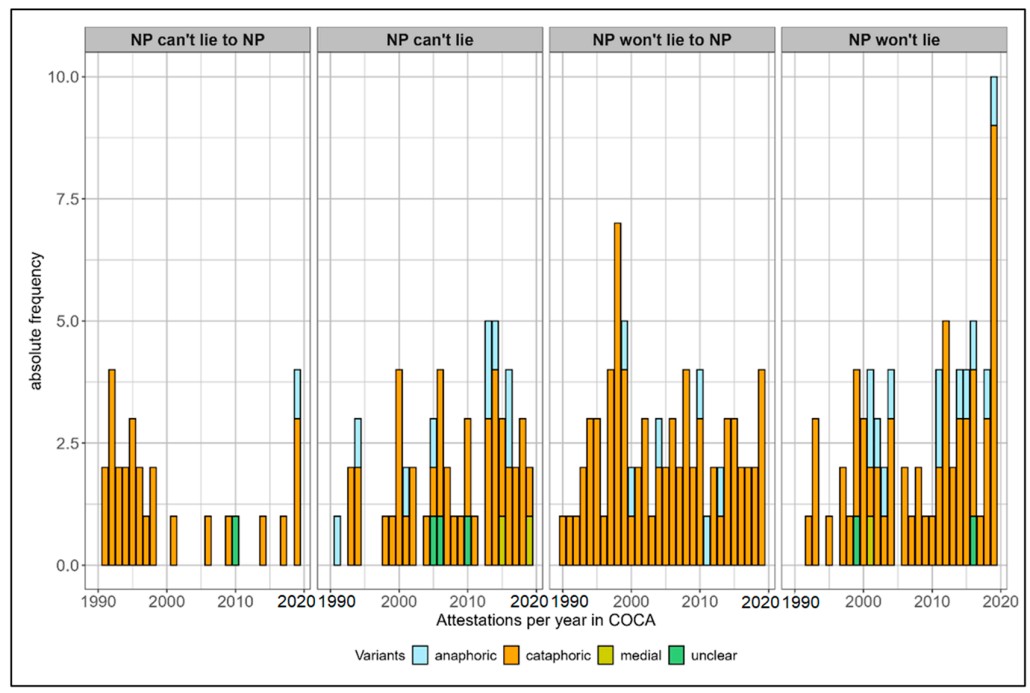

**Figure 7.** Results for reference/position analysis of *can't* and *won't* variants in COCA in absolute numbers. Anaphoric and cataphoric reference can be regarded as equivalent to right- and left-peripheral, while medial refers to clause-medial position.

Lastly, of the 153 attestations featuring *won't* variants, 131 have cataphoric reference (~86%), 19 have anaphoric reference (~12%), 1 is positioned medially (~1%) and 2 have unclear reference (~1%). *NP won't lie to NP* (77 attestations) has cataphoric reference 71 times (~92%) and anaphoric reference 6 times (~8%), cf. Figure 7 and Table 1. *NP won't lie* (74 attestations) has cataphoric reference 59 times (~79%), anaphoric reference 12 times (~16%), medial position 1 time (~1%) and unclear reference 2 times (~3%). The only attestation of *NP won't lie about NP* has anaphoric reference, and the only attestation of *won't lie* has cataphoric reference.

*3.3. Function*

After the previous section dealt with the formal development of *not gonna lie* and its variants, this section will detail the functional development, especially as it relates to FTA mitigation (Brown & Levinson, 1987). This analysis only takes into account non-literal attestations, i.e., attestations judged as either epistemic or ambiguous in meaning. This left 842 (of 1299) attestations for analysis.

Of these 842 non-literal attestations, 559 feature *going-to* in the modal slot (of 669; 84%). In total, 229 of these are instances of *NP BE not GOING TO lie to NP* (of 282 in total; ~81%), 8 of *NP BE not GOING TO lie about NP* (of 23; ~35%), 291 of *NP BE not GOING TO lie* (of 327; ~89%), 2 of *not GOING TO lie to NP* (100%) and 29 of *not GOING TO lie* (100%). There are no non-literal attestations of *NP BE not GOING TO lie for NP*. Moreover, the subject slot (where applicable) is exclusively filled with first-person pronouns (almost exclusively *I*, but there are 4 attestations featuring *we* across all variants). The object slot (where applicable) is almost exclusively filled with *you*, except for the *about-NP* variants, in which the object slot is occupied by either *it* or *that*.

In general, most of the non-literal *going-to* attestations occur in the context of a potential FTA. Only 29 of the 559 analyzed attestations cannot be categorized because of missing context (13 attestations) or do not occur in the context of an FTA (16 attestations). Overall, the *going-to* variants occur most commonly in the context of an FTA against the speaker's positive face, more specifically, confessions, such as the following:

7.  "**I'm not going to lie**," says former Louisiana Tech coach Leon Barmore, who kept tabs on the game while playing golf. "I was pulling for Tennessee, sure I was." (2003, NEWS).

8.  "The swimming, the swimming is a killer. **I'm not going to lie**. I'm not a natural swimmer." (2008, SPOK).

However, there are examples for each FTA category in the data. FTAs against the hearer's negative face are mostly warnings or advice (cf. 9), while FTAs against the hearer's positive face are often delivering bad news (cf. 10), the speaker bragging (cf. 11) or expressing criticism (cf. 12).

9.  "Now, **I'm not gonna lie to you**. The job comes with more responsibility, but it offers a lot more rewards." (2005, MOV).

10. "Look, Sam, **I'm not going to lie to you**. It's going to be a while before your brother wakes up" (2010, TV).

11. "Of my own company... **I'm not gonna lie to you**, I get a lot of perks." (2009, MOV).

12. "You—no, **I'm not going to lie**. You're pretty and all, but your (censored) [sic] attitude stinks" (1997, SPOK).

As a next step, the FTA analysis was conducted per variant (cf. Figure 8 and Table 2 for percentages per variant). *NP BE not GONG TO lie to NP* does follow the overall tendency of occurring in the context of FTAs against the speaker's positive face (91 attestations, ~40%) but also frequently occurs in the context of other FTAs: 54 attestations occur with FTAs against the hearer's negative face (~24%), 64 with FTAs against the hearer's positive face (~28%) and 4 with FTAs against the speaker's negative face. In 9 cases, no apparent FTA is committed, and 7 examples cannot be classified.

*NP BE not GOING TO lie* is much less variable in its function. It occurs in the context of FTAs committed against the speaker's positive face in 201 cases (~69%). In total, 47 attestations occur in the context of FTAs against the hearer's positive face (~16%) and 32 in the context of FTAs against the hearer's negative face (~11%). There are no instances of an FTA committed against the speaker's negative face. In 7 cases, no FTA is committed, and in 4 examples, the context is unclear and cannot be classified. *Not GOING TO lie* occurs in

the context of FTAs against the speaker's positive face 18 times (62%) and in the context of FTAs against the hearer's positive face 10 times (~34%). 1 example could not be classified.

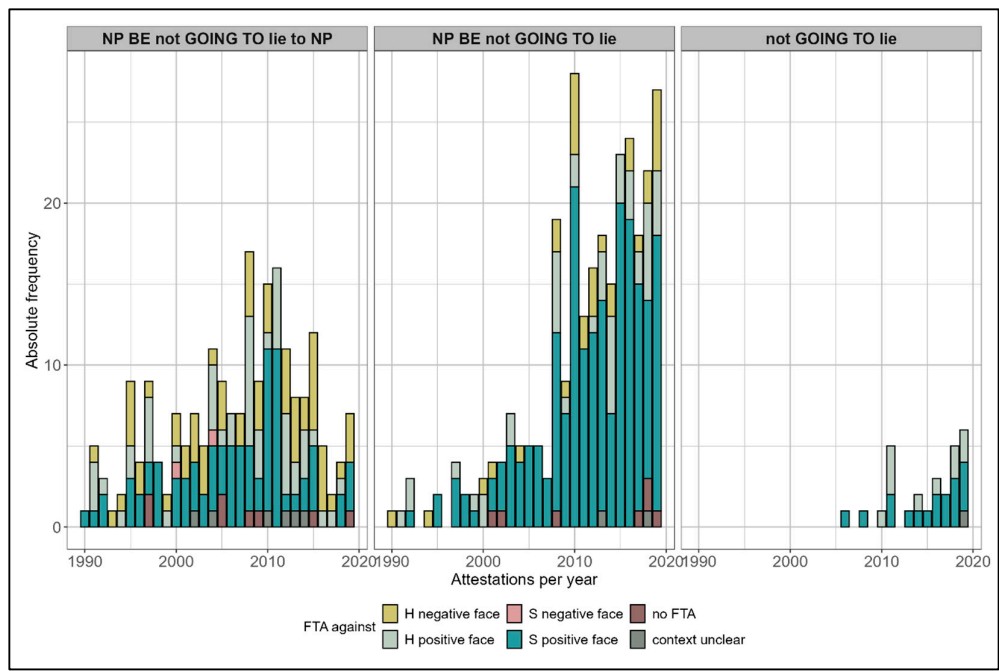

**Figure 8.** Results of the FTA analysis of going-to variants (non-literals uses) in COCA.

**Table 2.** Results of the FTA analysis in percent relative to the number of attestations of each variant.

| Variant | S pos. | H neg. | H pos. | S neg. | no FTA | Unclear |
|---|---|---|---|---|---|---|
| *NP BE not GOING TO lie to NP* | 40% | 28% | 24% | 2% | 4% | 3% |
| *NP BE not GOING TO lie about NP* | 50% | 13% | 25% | - | 13% | - |
| *NP BE not GOING TO lie* | 69% | 11% | 16% | - | 2% | 1% |
| *not GOING TO lie* | 62% | - | 34% | - | - | 4% |
| *NP can't lie to NP* | 67% | 10% | 13% | - | 3% | 7% |
| *NP can't lie* | 68% | 8% | 7% | - | 3% | 14% |
| *can't lie* | 100% | - | - | - | - | - |
| *NP won't lie to NP* | 53% | 19% | 26% | - | - | 1% |
| *NP won't lie about NP* | 100% | - | - | - | - | - |
| *NP won't lie* | 69% | 14% | 12% | 1% | - | 4% |
| *won't lie* | - | - | - | - | 100% | - |

The 8 attestations of *NP BE not GOING TO lie about NP* (not depicted in Figure 8) do not show a clear tendency because of the low frequency of the pattern. 4 examples occur in the context of an FTA committed against the speaker's positive face, 2 occur in the context of an FTA against the hearer's positive face and 1 in the context of an FTA against the hearer's negative face. The remaining attestation cannot be classified. Both attestations of *not GOING TO lie to NP* occur in the context of an FTA against the speaker's positive face (not depicted in Figure 8).

The same analyses were conducted for the *can't* and *won't* variants. There are 92 non-literal *can't* variants in the data (of the original 322), 30 of *NP can't lie to NP*, 59 of *NP can't lie*, 2 of *can't lie* and 1 of *can't lie to NP*. Of *NP can't lie for NP* and *NP can't lie about NP*, there are only literal attestations. Similar to the *going-to* variants, the subject slot (where applicable) is mostly filled with first-person pronouns, usually *I* (in 87 cases). There is 1 instance of *we* and then 1 ambiguous attestation with *you* in the subject slot. The object slot (where applicable) is always filled with *you*.

Most non-literal *can't* variants occur in the context of FTAs against the speaker's positive face (63 attestations or ~68%, cf. Figure 9), cf. (13), which constitutes a personal confession; 8 attestations each occur in the context of FTAs against the hearer's negative and positive face, cf. e.g., (14), in which a negative emotion about the H is expressed, and (15), in which the S gives good news about themselves.

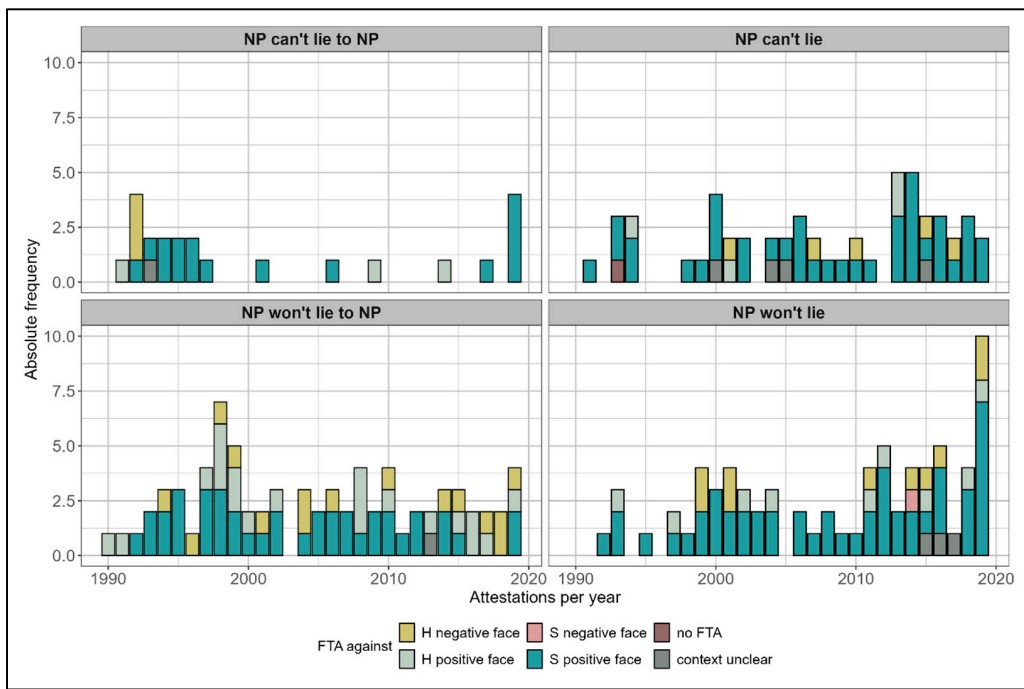

**Figure 9.** Results for FTA analysis of can't and won't variants (non-literal uses) in COCA.

13. "**I can't lie**. I'm missing my little brother every minute of every day" (2008, TV).
14. "When I left, **I can't lie**, I resented you." (2015, TV).
15. "I've never been this comfortable with myself before. And **I can't lie to you**; I'm happier than ever." (2014, MAG).

In 10 cases, the context did not allow for categorization, and in 3 cases, there was no FTA in the immediate context of the *can't* marker. 20 attestations of *NP can't lie to NP* (~67%) occur in the context of FTAs against the speaker's positive face, 3 against the hearer's negative face and 4 against the hearer's positive face (cf. Figure 7). In 1 case, no FTA is committed, and in 2 cases, the context is unclear. For *NP can't lie*, 40 attestations (~68%) occur in the context of an FTA committed against the speaker's positive face, 5 against the hearer's negative face and 4 against the hearer's positive face. In 2 cases, no FTA is committed, and in 8 cases, the context is unclear. Both attestations of *can't lie* and the one attestation of *can't lie to NP* occur in the context of an FTA committed against the speaker's positive face.

The *won't* variants are used non-literally 153 times in the data (of 212 attestations in total), 77 of *NP won't lie to NP*, 74 of *NP won't lie*, 1 of *NP won't lie about NP* and 1 of *won't lie*. There are only literal attestations of *NP won't lie for NP*. Like for the other variants, the subject slot of the non-literal attestations is filled with first-person pronouns (*I* in 150 cases and *we* in 2 cases) and the object slot with *you* (for all *to-NP* attestations) and *that* (for the *about-NP* variant).

Like the other variants, the *won't* variants are mostly used in the context of FTAs against the speaker's positive face (93 of 153 attestations; ~61%), cf. (16), which constitutes a confession. In total, 25 attestations are used in the context of FTAs against the hearer's negative face, cf. (17), in which a warning is expressed, 29 against the hearer's positive face,

cf. (18), in which criticism against the H is expressed, and 1 against the speaker's negative face, cf. (19), in which the S makes excuses.

16. "**I won't lie**," she says. "I just didn't have the patience." (1995, NEWS).
17. "**I won't lie to you**, Neo. Every single man or woman who has fought an agent has died." (1999, MOV).
18. "Boy, I got to tell you, **I won't lie to you**. It's the biggest screwup ever to go through this office." (2017, MOV).
19. "My Dad made me, **I won't lie**." (2014, TV).

In 1 case, no FTA is committed, and 4 attestations cannot be classified. In total, 41 attestations (of 77) of *NP won't lie to NP* occur in the context of an FTA committed against the speaker's positive face (~53%), 15 against the hearer's negative face (~19%) and 20 against the hearer's positive face (~26%). 1 attestation cannot be classified. Of the 74 attestations of *NP won't lie*, 51 occur in the context of an FTA committed against the speaker's positive face (~69%), 10 against the hearer's negative face (~14%), 9 against the hearer's positive face (12%) and 1 against the speaker's negative face. 3 attestations cannot be classified. The only attestation of *won't lie* occurs in a context in which no apparent FTA is committed, and the only attestation of *NP won't lie about NP* occurs in the context of an FTA against the speaker's positive face.

## 4. Discussion

This paper aimed to trace the development of the pragmatic marker *not gonna lie* in COCA. The results showed that there are many possible variants of the pattern *NP BE neg MODAL to lie PREP NP* and also showed that this pattern becomes more frequent over time. However, most variants have a very low token count. Several of the higher-frequency variants were investigated in more detail: *NP BE not GOING TO lie to NP*, *NP BE not GOING TO lie*, *not GOING TO lie*, *NP won't lie to NP*, *NP won't lie*, *NP can't lie to NP* and *NP can't lie*. Of these, mostly *NP BE not GOING TO lie* and *not GOING TO lie* increase in frequency over time. While there is some degree of variability in all open slots, only those variants with *I* in the subject slot (and very rarely *we*) and *you* in the object slot are used epistemically and become usualized. The negated modal remains variable, even though there is a strong preference for *not going to*, but *can't* and *won't* can be used as well. This is likely because the modal is not as discursively important as the subject and object. Furthermore, the longer variants, i.e., the variants with the prepositional phrase, are more variable than the shorter variants. Inversely, the shorter variants are more positionally flexible than the longer ones. The *going-to* variants are the least variable overall, while the *can't* variants are the most variable. The *going-to* variants are also the variant group that is most often used epistemically, while the *can't* variants are mostly used literally. Functionally, the epistemic variants are mostly used to mitigate attacks against the speaker's positive face in the context of confessions.

Viewed through the lens of the EC-Model, this means that through contextualization, the entire phrase becomes more and more associated with this mitigating function over time and is no longer used literally to refer to actual lies, which suggests semantic bleaching of component parts, i.e., contextual symbolization. This contextual symbolization likely leads to paradigmaticalization: Because of this association of the pragmatic marker *(I'm) not gonna lie* with this mitigating function in the context of FTAs against the speaker's positive face, the marker likely enters into competition with other pragmatic markers with similar functions. What this competition looks like in detail was not investigated empirically in the present paper, but the previous literature suggests mitigation seems to be a general function of TRUTH markers, e.g., *frankly* is used to mitigate face attacks, but rather in contexts of disagreements (Tseronis, 2011, p. 483) or unexpected information (Keizer, 2018, p. 69),

i.e., threats against the hearer's positive face (Brown & Levinson, 1987, p. 66). The fact that (some) pragmatic markers with source lexemes from the domain of TRUTH/FACT seem to gravitate towards a mitigating function might point towards a phenomenon that Hansen (2018) calls "onomasiological cyclicity", where a similar pragmatic function "is renewed several times by etymologically unrelated forms with similar content-level source meanings" (Hansen, 2018, p. 64); however, this needs further exploration.

While some decrease in variability in the subject slots for shorter variants suggests "deparadigmaticalization" of individual elements during the usualization process, the overall variability of slot fillers, especially in the modal slot, remains high. This has problematic implications for the syntagmaticalization of *(I'm) not gonna lie*. Some degree of decrease in external syntagmatic conformity can be observed by the omission of elements. This mostly affects the prepositional phrase, which is also reflected in the decrease in transitional probabilities from *I AM not GOING TO lie* to *to*. The later attestations also omit the subject, although these ungrammatical forms (at least in standard American English) are quite rare compared to the grammatical ones, which, in fact, suggests retention of external syntagmatic conformity. The marker also becomes more positionally flexible over time and is not only used with cataphoric but also with anaphoric reference. It is rarely placed medially, however. In terms of internal syntagmatic conformity, it is true that the phrase becomes more fixed and that the phrase is only used as a pragmatic marker when *I* is in the subject slot. The strong preference for *going to* in the modal slot and the slight increase in the transition probabilities from *I AM not GOING TO* to *lie* further suggest some degree of syntagmaticalization. It can, however, not be claimed that earlier elements become good predictors for later elements, as is the case for *needless* in the comment clause *needless to say*, as investigated in Schmid (2020). All of the elements in *(I'm) not gonna lie (to you)* are quite common either by themselves or in combination, so a stark increase in transition probabilities over time likely only happens in pragmatic markers with less frequent items.

The fact that modal variation remains through the entire time frame and that variants with other modals do not decrease in use (and, for the epistemic uses, seem to, in fact, become more popular), even after *(I'm) not gonna lie* rapidly increases in frequency, also complicate the matter, as more advanced syntagmaticalization should mean less variation within the syntagmatic string. It is possible that this variation is due to regional variability or due to other extralinguistic variables that cannot be investigated because of the lack of metadata available for the corpus. It is also possible that because all the elements of the source clause are so frequent outside of this pragmatic marker, this frequency prevents the individual elements from becoming good predictors for later elements, which then, in turn, allows for more variability. It could also be the case that the variability in the modal remains because the modal is the least important element for the function of the marker. The only two elements that are important for the invited inference that arises through invoking Grice's Maxim of Quality and, therefore, flouting the Maxim of Quantity (as suggested by Lenker, 2010; Niehüser, 1987; Zeschel et al., 2025) are the verb *lie* and its negation. Subject *I* and object *you* are important for their suggestion of intersubjectivity, although they are discursively optional because references to the addresser and addressee are implied in communicative situations and thus can be omitted, which, in part, mirrors what Berry (2018) posits for the development of *frankly*. The modal does not seem to add any type of metacommunicative functionality to the pragmatic marker, which might be why this slot remains more variable than expected. It is also perfectly possible that in more recent data, this variation no longer occurs to this degree.

## 5. Conclusions

In summation, the goal of the present paper was to investigate the development of the pragmatic marker *not gonna lie* from a formal and functional perspective and to evaluate how the Entrenchment-and-Conventionalization Model can be applied to describe such a process. While it could be shown that usualization is definitely suitable (even though the application of syntagmaticalization has proven difficult), only one subsection of the EC-Model was explored, and even that one not fully. Since the present paper only looked at one pragmatic marker and its variants in detail, paradigmaticalization, i.e., how existing pragmatic markers with similar functions react to the emergence of a new one, was not investigated. Furthermore, the second part of conventionalization, diffusion, i.e., how new forms spread through different registers and communities, was not explored. While COCA is an extremely large dataset that covers a larger timeframe than most other corpora of contemporary American English, it does not allow for sociolinguistic investigations. Lastly, entrenchment, i.e., how the language of individual speakers is influenced by, and themselves influence, language use and structure, was not explored in this study. As an outlook, the limited scope of the present study therefore opens several avenues for exciting future research on the diffusion of pragmatic markers, for example, in online spaces, as was, e.g., conducted in Kerremans et al. (2018) or Würschinger (2021) for neologisms, or how emerging pragmatic markers become routinized in the minds on individuals, as was conducted within a grammaticalization framework for the *let alone* construction in Neels (2020).

**Funding:** This research received no external funding.

**Institutional Review Board Statement:** Not applicable.

**Informed Consent Statement:** Not applicable.

**Data Availability Statement:** No new data was created or analyzed in this study. Data sharing is not applicable to this article.

**Conflicts of Interest:** The author declares no conflicts of interest.

## Abbreviations

The following abbreviations are used in this manuscript:

| | |
|---|---|
| COCA | Corpus of Contemporary American English |
| NP | Noun phrase |
| PREP | Preposition |
| FTA | Face-threatening act |
| H | Hearer |
| S | Speaker |

## Appendix A

**Table A1.** List of the lower-frequency variants found in COCA and their absolute frequency.

| Variant | Absolute Frequency |
|---|---|
| *NP cannot tell a lie* | 39 |
| *NP BE not GOING TO lie about NP* | 23 |
| *NP will not lie* | 20 |
| *NP can't lie about NP* | 19 |
| *NP will not lie to NP* | 16 |
| *NP cannot lie* | 10 |

Table A1. *Cont.*

| Variant | Absolute Frequency |
|---|---|
| *NP won't lie about NP* | 8 |
| *NP won't lie for NP* | 8 |
| *NP cannot lie to NP* | 6 |
| *NP not GOING TO lie for NP* | 6 |
| *can't lie to NP* | 4 |
| *not GOING TO lie to NP* | 2 |
| *NP can't lie for NP* | 2 |
| *NP will not lie about NP* | 2 |
| *can't lie* | 2 |
| *no lie* | 1 |
| *NP cannot tell a lie to NP* | 1 |
| *NP will not tell a lie* | 1 |
| *won't lie* | 1 |

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
