# Peer review of "“Not gonna lie, that’s a real bummer”—The Usualization of the Pragmatic Marker not gonna lie"

_languages, doi:10.3390/languages10080186_

Round 1

Reviewer 1 Report

Comments and Suggestions for Authors

I like this paper and think it has the potential to make a valuable contribution. My comments are intended to help the authors focus and strengthen the work.

  • Example (2): This is a perfectly fine example but do we really need to include it? Snuff films centre on horrific sexualized violence against women. Surely you can find an equally viable example that is not awful.
  • Is 'not gonna lie' already established in the literature as a pragmatic marker? The paper takes it as a given that it is. If that is the case, then the paper would be strengthened by incorporating/citing work discussing/establishing its status.
  • Why COCA? We are told it is used and why certain parts are excluded, but the choice itself is never motivated. It would be helpful to do so, relative to the aims of this study. This is especially important given the constraints of the corpus, which affect what you can and can't achieve in the analysis.
  • On page 6 you differentiate between literal and epistemic meanings, but these are not explained and exemplified. It would be helpful if you were to do so.
  • At the start of the paper, you say you perform both a qualitative and a quantitative analysis, yet section 3 (results) does not map to this in any direct way. Rather than talking about "parts" of the results, could you reframe this in terms of the methods, goals and research questions in a more transparent way?
  • Figure 1 is impossible to read, and will not be accessible once in print. The information would be better presented as a table or as a line graph. It's not clear to me what is gained by the current level of granularity, beyond absolute transparency about the constituency of your data over the sample time period. Could you present the key points here and provide the full information in an appendix? You clearly have a steep A-curve distributionally, which is expected.
  • Sorry, I know you mentioned it briefly in the introduction but I thought this paper was about 'not going to lie'. Why does the analysis include other forms? How do they inform us of what is going on with 'not going to lie'? It would be helpful if the relevance were made clear in the background/contextualization of the study. At the moment, it feels like a distraction in the story you are building.
  • In fact, I'm not sure why the focus isn't simply the pragmatic uses of 'not going to lie'. Why are the literal ones included?
  • Figures 2 and 3: Again, I'm not sure I understand why you are opting for stacked bar graphs rather than line graphs here. They actually make it harder to discern what is going on with individual forms over time, because they obfuscate linear (temporal) patterns.
  • There are a number of short sections in this paper, and the graphs are too busy. This makes reading the paper more challenging, because the narrative is constantly broken up and disrupted. I think the paper would benefit from stepping back, assessing what the key research question is, and then focusing on that. I think there's a lot here that can be removed to form the core of a different paper.
  • I would combine sections 4 and 5. 

Author Response

Thank you for taking for engaging with my article and for your extensive feedback.

Comment 1: Example (2): This is a perfectly fine example but do we really need to include it? Snuff films centre on horrific sexualized violence against women. Surely you can find an equally viable example that is not awful.

Response: Yes, I see your point regarding the potentially triggering nature of the original example. I was trying to find examples that show the mitigating function of not gonna lie without the need for much context, but I have now chosen a different example that should illustrate my point just as well (cf. line 62f of the manuscript).

Comment 2: Is 'not gonna lie' already established in the literature as a pragmatic marker? The paper takes it as a given that it is. If that is the case, then the paper would be strengthened by incorporating/citing work discussing/establishing its status.

Response: As mentioned in l. 186f., there is no previous literature on not gonna lie that I am aware of. In my opinion, the work on similar markers in both English and German (I must admit by Brinton 2017, ehrlich gesagt & um die Wahrheit zu sagen by Niehüser 1987, Rolf 1994, Hagemann 1997 and Zeschel et al. 2025), the crowd-sourced online definitions that identify not gonna lie as synonymous with more established and prototypical pragmatic markers e.g., honestly, the multiple usage examples and the functional analysis that established the clear pragmatic function of not gonna lie should be sufficient to claim that it is, in fact, a pragmatic marker.

Comment 3: Why COCA? We are told it is used and why certain parts are excluded, but the choice itself is never motivated. It would be helpful to do so, relative to the aims of this study. This is especially important given the constraints of the corpus, which affect what you can and can't achieve in the analysis.

Response: That you for this very helpful comment. I have added the motivation behind the choice to section 2. of the manuscript (l. 255ff.). not gonna lie is rather infrequent, which means that a large dataset is required and COCA is among the biggest. It is of course true that COCA is largely standard American English and a corpus with more natural spoken data or online data would be a much better fit, however, I am not aware of any corpora of spoken American English that have the necessary size or cover the relevant time frame (1980/1990s to today), which is why I chose COCA for this study despite its shortcomings.

Comment 4:  On page 6 you differentiate between literal and epistemic meanings, but these are not explained and exemplified. It would be helpful if you were to do so.

Response: I have added an explanation for both terms along with references to studies that use these terms in a similar fashion (l. 283f.)

Comment 5: At the start of the paper, you say you perform both a qualitative and a quantitative analysis, yet section 3 (results) does not map to this in any direct way. Rather than talking about "parts" of the results, could you reframe this in terms of the methods, goals and research questions in a more transparent way?

Response: I have rephrased the section in question to reflect the expectations/hypotheses laid out in the introduction (l. 370ff. of the manuscript). The formal analysis constitutes an investigation into syntagmaticalization and deparadigmaticalization of the component parts of the markers, as does the positional analysis. I have therefore switched the functional and the positional analysis, so that all analyses into syntagmaticalization are more closely connected in the results section.

Comment 6 : Figure 1 is impossible to read, and will not be accessible once in print. The information would be better presented as a table or as a line graph. It's not clear to me what is gained by the current level of granularity, beyond absolute transparency about the constituency of your data over the sample time period. Could you present the key points here and provide the full information in an appendix? You clearly have a steep A-curve distributionally, which is expected.

Comment 7: Figures 2 and 3: Again, I'm not sure I understand why you are opting for stacked bar graphs rather than line graphs here. They actually make it harder to discern what is going on with individual forms over time, because they obfuscate linear (temporal) patterns.

Response: Since comment 6 & 7 mention similar issues, I would like to respond to them together. Yes, I agree that Figure 1 is too busy, which is why I have removed the lower-frequency variants from the figure, which are now listed in Appendix A. I have opted to use stacked bar graphs as opposed to line graphs because the same information in line graphs is actually much more difficult to discern. The short time frame and the general low-frequency and variability of the individual variants make the line graphs incredibly busy and smoothing them removed a lot of information since the standard deviation is quite high. The only two variants and meaningfully increase over time are I'm not gonna lie and not gonna lie which is clearly visible in Figure 2. Figure 3 is meant to illustrate that these variants do not increase (as opposed to the going-to variants) and I believe the figure does serve this purpose. 

Comment 8: Sorry, I know you mentioned it briefly in the introduction but I thought this paper was about 'not going to lie'. Why does the analysis include other forms? How do they inform us of what is going on with 'not going to lie'? It would be helpful if the relevance were made clear in the background/contextualization of the study. At the moment, it feels like a distraction in the story you are building.

Response: The analysis includes this other forms because most of them are also used in the same/similar function to not gonna lie, especially the won't and can't variants (cf. also example 5 in line 342) and I believe their inclusion makes sense based on the results of the functional analysis. in light of how the effects of syntagmaticalization, i.e., loss of paradigmatic choices within the expression that is in the process of usualizing, are laid out in Schmid (2020), this continuing variability is quite unexpected. In order to make this more clear, I have added additional examples in section 3.3.2 (lines 661ff. and l. 688ff.).

Comment 9: In fact, I'm not sure why the focus isn't simply the pragmatic uses of 'not going to lie'. Why are the literal ones included?

Response: The idea of including the literal uses is based on the idea of semantic bleaching. If string "I am not going to lie to you" is always used pragmatically, does it then make sense to claim that the meaning of "lie" was bleached? The result that the strings with "can't" are most often used literally and the ones with "won't" at least to a lesser degree than the ones with "going to", supports the choice to include the literal ones in my opinion because it shows that some strings are more susceptible to undergo usualization than others.

Comment 10: There are a number of short sections in this paper, and the graphs are too busy. This makes reading the paper more challenging, because the narrative is constantly broken up and disrupted. I think the paper would benefit from stepping back, assessing what the key research question is, and then focusing on that. I think there's a lot here that can be removed to form the core of a different paper.

Response: I am afraid that I do not agree here, since in my opinion, all the information presented is relevant in order to investigate the development of not gonna lie in the context of the EC-Model. However, I appreciate your point about the short sections. In order to make the results section less broken up, I have moved the graphs to the end of the respective sections, which makes them less disruptive to the flow of the text.

Comment 11: I would combine sections 4 and 5. 

Response: I prefer keeping the two section separate because section 5 presents more of an outlook and thus for me does not belong with the discussion.

Reviewer 2 Report

Comments and Suggestions for Authors

I think this is a very nice paper investigating developments in the use of the not gonna lie marker, and would recommend that it be published pretty much as is, although I do have some suggestions for minor amendments that the authors may want to consider.

The introduction and review of the field is very good, clearly motivates Schmid’s EC-model as an appropriate methodology for the work, and provides a high level of detail for others to replicate and verify the approach. The predictions of the EC-model with respect to the development of not gonna lie are clearly set out, the analysis is thorough and clearly explained, and the conclusions are justified and sensible.

Most of my suggestions relate to the presentation of the results. I found Figure 1 rather busy, though on reflection perhaps it was justified as a summary which leads into the more detailed analysis. I wonder whether it might be better to only include the more frequent variants on the figure (and an ‘others’ category), and have the specific detail in a table or an appendix.

At the end of section 3.1.2, perhaps the claim that forms omitting the prepositional phrase are less variable in the subject slot than the longer forms with prepositional phrase, regardless of the modal should be more carefully phrased, given that the subject slot in NP BE not GOING TO lie is filled with I in 96% of cases both with the PP and without.

I liked Figure 4 very much, but I’m not sure exactly what the authors are trying to conclude from the dip in transition probabilities I AM NOT GOING TO > lie; surely the preceding transition probabilities in the two alternatives are so low that no dip could occur.

At the end of section 3.1.3, the authors could consider a brief statement confirming that the analysis shows no clear evidence for the expected syntagmaticalization. This is covered in the final discussion, but I think it would be useful at the relevant point in the analysis too.

I also found a couple of typos:

P9 l403: lots -> slots

P18 l696: negMODAL needs a space.

Author Response

Thank your for your thoughtful comments and constructive feedback.

Comment 1: Most of my suggestions relate to the presentation of the results. I found Figure 1 rather busy, though on reflection perhaps it was justified as a summary which leads into the more detailed analysis. I wonder whether it might be better to only include the more frequent variants on the figure (and an ‘others’ category), and have the specific detail in a table or an appendix.

Response: Thank you for the suggestion, I agree and have adjusted Figure 1 accordingly.

Comment 2: At the end of section 3.1.2, perhaps the claim that forms omitting the prepositional phrase are less variable in the subject slot than the longer forms with prepositional phrase, regardless of the modal should be more carefully phrased, given that the subject slot in NP BE not GOING TO lie is filled with in 96% of cases both with the PP and without.

Response: Agreed, thank you for the comment. I have rephrased this claim to clarify that the decrease in variability only applies to the variants with can't and won't (cf. lines 470 of the manuscript).

Comment 3: I liked Figure 4 very much, but I’m not sure exactly what the authors are trying to conclude from the dip in transition probabilities I AM NOT GOING TO > lie; surely the preceding transition probabilities in the two alternatives are so low that no dip could occur.

Response: The suggestion here is that I AM not GOING TO is a relatively common, expected string, but that lie after I AM not GOING TO is unexpected. I followed by won't is in general more unexpected than I > AM not GOING TO but I won't > lie is actually more likely than I > won't. However, I agree that for the overall interpretation and research question, this is an unnecessary side track that not picked up later at all, which is why I deleted it.

Comment 3: At the end of section 3.1.3, the authors could consider a brief statement confirming that the analysis shows no clear evidence for the expected syntagmaticalization. This is covered in the final discussion, but I think it would be useful at the relevant point in the analysis too.

Response: Yes, I agree, I have added a comment regarding the lack of evidence for increased internal syntagmatic conformity from the transition probabilities (cf. l. 512 of the manuscript)

Comment 4: I also found a couple of typos:

Response: I have corrected these typos.